# Quantifying the Gain in Weak-to-Strong Generalization

**Moses Charikar**
Stanford University
moses@cs.stanford.edu

**Chirag Pabbaraju**
Stanford University
cpabbara@cs.stanford.edu

**Kirankumar Shiragur**
Microsoft Research
kshiragur@microsoft.com

## Abstract

Recent advances in large language models have shown capabilities that are extraordinary and near-superhuman. These models operate with such complexity that reliably evaluating and aligning them proves challenging for humans. This leads to the natural question: can guidance from weak models (like humans) adequately direct the capabilities of strong models? In a recent and somewhat surprising work, Burns et al. [BIK+23] empirically demonstrated that when strong models (like GPT-4) are finetuned using labels generated by weak supervisors (like GPT-2), the strong models outperform their weaker counterparts—a phenomenon they term *weak-to-strong generalization*.

In this work, we present a theoretical framework for understanding weak-to-strong generalization. Specifically, we show that the improvement in performance achieved by strong models over their weaker counterparts is quantified by the *misfit error* incurred by the strong model on labels generated by the weaker model. Our theory reveals several curious algorithmic insights. For instance, we can predict the amount by which the strong model will improve over the weak model, and also choose among different weak models to train the strong model, based on its misfit error. We validate our theoretical findings through various empirical assessments.

## 1 Introduction

Present-day AI models demonstrate incredible capabilities at a variety of extremely difficult tasks. For this reason, they are frequently described as being *superhuman*, in that it seems hard to imagine a human displaying the same abilities as the AI model. For example, move 37 in AlphaGo's famous victory against Go expert Lee Sedol [Met16] has been described as being beyond the realm of human imagination. In this sense, today's AI models are well on the path of exhibiting *new* and *emergent* abilities [WTB+22]. Ultimately, we want these new abilities to be aligned with what would be beneficial to humanity. This rationale is what primarily guides the training of large-scale AI models through human feedback [CLB+17]. However, given that we expect AI models to pick up skills that we ourselves don't fully grasp as humans, how can we enable these highly capable models to realize their potential?

A recent work by [BIK+23] shows that not all hope is lost in this endeavor. To model humans as being *weak* supervisors for increasingly *strong* AI models, they conduct the following "weak-to-strong generalization" experiment. Suppose we finetune a small language model like GPT-2 [RWC+19] on data with ground-truth labels for a task. What happens if we then finetune a large language model like GPT-4 [Ope23a] on data labeled by GPT-2, instead of data having ground-truth labels? Would GPT-4 simply overfit to GPT-2's labels and do no better, or would it outperform GPT-2, given that it is inherently a much stronger model? The surprising experimental result is that GPT-4 trained in this manner outperforms GPT-2 when evaluated on the true data, for a variety of finetuning tasks. Note

that GPT-4 is able to outperform GPT-2 without ever seeing true labels when it was being finetuned. One plausible explanation for this is that GPT-4 was able to glean the essence of the finetuning task from GPT-2's labels, and since it is fundamentally a stronger model than GPT-2, this knowledge was sufficient for it to outperform GPT-2.[1]

In this work, we seek theoretical justification for why we might expect to see such a gain in accuracy in weak-to-strong generalization. Concretely, we ask:

> *Does a weakly supervised strong model provably attain smaller error than its weak supervisor, and if so, can this gain be formally quantified?*

Towards answering this question, we show (Theorem 1) that in the concrete setting of regression, the true error of a strong model trained on weak labels is smaller than the error of the weak model, by *at least* the error of the strong model on the weak labels itself. We call this latter quantity the *misfit* between the weak and strong model. Our result can be stated as the following simple principle:

Gain in accuracy in weak-to-strong generalization $\approx$ Misfit between the weak and strong model

Intuitively, the misfit quantifies the *erroneous knowledge* that the strong model *does not* obtain from the weak model, and hence also the amount that the strong model *improves* over the weak model. We note that the work of [BIK$^+$23] does empirically show that the performance gain of the strong model scales directly with its misfit (or *disagreement*) with the weak model; our result thereby provides a precise quantification of this observation.

Key to obtaining our results is a *representation-theoretic perspective* [TJJ20] towards weak-to-strong generalization. We posit that the main difference between weak and strong models is in the disparity between the quality of their data representations. This disparity in representation quality can manifest, among other reasons, due to a difference in the expressivity and complexity of the weak and strong models, and the amount of pretraining data that they have seen. For example, in the experiments by [BIK$^+$23], the weak and strong models used are GPT-2 and GPT-4 respectively; the latter is a significantly larger transformer architecture, pretrained on a much larger dataset than the former. As a broader analogy, consider the task of learning a new language. This is an easier task for a multilingual person than a monolingual person. A multilingual person has a richer representation for language, drawing from their knowledge of different syntax, lexical structures, and sounds in multiple languages. With this perspective, we can imagine finetuning tasks to be relatively simple functions (e.g., linear functions) composed with the appropriate representation. For example, if the task is about learning Italian, a suitable "Italian-specific" linear combination of the multilingual's representation of the problem (including features learned from Spanish and French, say) might allow them to better understand the new language, while the same might not work so well for a monolingual whose representation only has features learned from English.

Armed with this perspective, we model the task of learning a real-valued finetuning task under the least squares loss in the weak-to-strong generalization framework. We assume that there exists a ground-truth representation $h^\star$ of the data, which makes it amenable to learn a finetuning task $f^\star$ of interest. We imagine that the weak and strong models come equipped with representation maps $h_w$ and $h_s$ respectively, which are possibly obtained via pretraining on a corpus of data. Next, we imagine that the weak model sees data labeled by the target function $f^\star \circ h^\star$, and after finetuning, learns some arbitrary function $f_w \circ h_w$. At this point, the weak supervision pipeline begins. The strong model is fed with data labeled by $f_w \circ h_w$ (instead of the true labels $f^\star \circ h^*$), and as part of finetuning, outputs a function $f_{sw}$ from a function class $\mathcal{F}_s$, that minimizes the discrepancy between $f_{sw} \circ h_s$ and the data labeled by $f_w \circ h_w$ that it sees. Ultimately, we care about the error of $f_{sw} \circ h_s$ with respect to the *true* finetuning task, namely $f^\star \circ h^\star$. Our main result (Theorem 1) precisely quantifies the gain in the accuracy of $f_{sw} \circ h_s$ over $f_w \circ h_w$ in terms of the misfit between them, under the assumption that the set of functions $\mathcal{F}_s$ is a *convex set*.[2] In many practical applications, the representation map is generally the forward pass of the data through a suitable neural network architecture, and the finetuning task is performed by the *last linear layer* [KRJ$^+$22] of the network. In such cases, our assumption that the set $\mathcal{F}_s$ is convex readily holds true.

---

[1]We note however that this methodology did not allow GPT-4 to fully reach its performance achieved by training on true data.

[2]Note that the functions in $\mathcal{F}_s$ need not be convex themselves.

We validate our characterization of the gain in weak-to-strong generalization through various experiments (Section 5) on synthetic and real-world data. The experiments corroborate our theoretical findings. Namely, we observe that upon performing the above weak-to-strong supervision pipeline, the gain in accuracy of the weakly-supervised strong model over its weak supervisor more or less *exactly* aligns with the misfit between the weak and strong models (Figure 2). We also demonstrate (Section 5.3) that the labels "weak" and "strong" models are nuanced and not solely dependent on expressive power; in fact, in a low-sample regime, a less expressive model produces a higher quality representation and should be considered a strong model. Our theory and experiments lead to several algorithmic insights and open up interesting questions. For example, one algorithmic heuristic that arises from our theory is the following: given access to different weak models, choose to deploy the strong model that achieves the smallest difference between the weak model error and misfit (Table 1). Our results also motivate the perhaps counterintuitive algorithmic question of obtaining weak models that lead to *large misfits* with the strong model.[3] Another possible line of inquiry could look into *ensembling* across different weak models, and obtaining a gain close to the *sum* of their individual misfits. At a more philosophical level, this is akin to a superhuman AI model assimilating knowledge from various humans, while correctly identifying and discarding each of their flaws.

## 2 Related Work and Preliminaries

### 2.1 Related Work

The idea of converting a "weak" learner to a "strong" learner can be traced all the way back to the famous paradigm of *boosting* [Fre95, FS97], if not earlier. The recent work by [BIK+23] frames this problem within the context of *superalignment* [Ope23b, JQC+23] which seeks to reliably align AI models smarter than humans to human intent. Thereafter, several works that study the training of a "strong" model guided in some capacity by a "weak" model have emerged. Some of these include instruction filtering by weak models [LZH+24], easy-to-hard generalization [SYS+24], weak-to-strong correction [JCL+24] and weak-to-strong hallucination inducement [ZCBS23].

The weak-to-strong generalization paradigm is perhaps most closely related to the teacher-student model of training [LA16, TV17] (sometimes also referred to as knowledge distillation [HVD15, GYMT21]), where a student model (typically smaller) is trained using data labeled by a teacher model (typically larger), and possibly some additional ground-truth data. The remarkable phenomenon of the student model outperforming the teacher has been observed in many works [BCNM06, HVD15, FLT+18]. Most relevant to us are formulations where the student model is *equally* [FLT+18] or *more powerful* [XLHL20] than the teacher model. There has been theoretical work explaining superior generalization of the student in some specialized settings, e.g., the work of [MFB20] where the student also has access to ground-truth labels, or the work of [WSCM20], which operates under a certain *expansion* criterion and *consistency* loss. In contrast, our work does not assume that the student model has any access to ground-truth labels, and also does not incorporate a regularization term in the objective. Finally, the conceptual insight in our work about the performance gain of the student model scaling with its disagreement with the teacher model is closely related to the theory of generalization bounds based on disagreement between different classifiers [DLM01, WZ17, YHY+19].

### 2.2 Preliminaries

We assume that the data domain is $\mathbb{R}^d$, and assume that there exists a ground truth representation function $h^\star : \mathbb{R}^d \to \mathbb{R}^{d^\star}$ that maps the data $x$ to an enriched representation $h^\star(x)$. We assume the existence of pretraining tasks, through which strong models obtain representations of the data from a function class $\mathcal{H}_s : \mathbb{R}^d \to \mathbb{R}^{d_s}$, and weak models obtain representations from a function class $\mathcal{H}_w : \mathbb{R}^d \to \mathbb{R}^{d_w}$. For example, $\mathcal{H}_s$ can be the class of deep neural networks, and $\mathcal{H}_w$ can be the class of shallow neural networks. The target finetuning task (composed with the ground truth representation) is denoted as $f^\star \circ h^\star$, and the function learnt by the weak model is denoted by $f_w \circ h_w$. We assume that the strong model learns finetuning tasks from a function class $\mathcal{F}_s : \mathbb{R}^{d_s} \to \mathbb{R}$, and assume that the set $\mathcal{F}_s$ is a *convex* set. The convexity assumption requires that, for any $f, g \in \mathcal{F}_s$, and for any $\lambda \in [0, 1]$, there exists $h \in \mathcal{F}_s$ such that for all $z \in \mathbb{R}^{d_s}$, $h(z) = \lambda f(z) + (1 - \lambda)g(z)$. For example,

---

[3]In other words, weak models that largely misfit strong models are the ones that most effectively *elicit* [CCX22] what the strong models already know.

$\mathcal{F}_s$ can be the class of all linear functions from $\mathbb{R}^{d_s}$ to $\mathbb{R}$. However, we do not assume anything about either $f^\star$ or $f_w$; in particular, they need not belong to $\mathcal{F}_s$. We denote the marginal data distribution by $\mathcal{P}$. For any two functions $f, g : \mathbb{R}^d \to \mathbb{R}$, we define the distance $d_\mathcal{P}(f, g) = \mathbb{E}_{x \sim \mathcal{P}}(f(x) - g(x))^2$, i.e., it is the average (with respect to $\mathcal{P}$) squared distance between the images of the functions.

## 3 Results

We first state a quantitative version of our main result that characterizes the gain in weak-to-strong generalization in terms of strong-to-weak misfit in the so-called *realizable* setting. Namely, we assume that the target finetuning task $f^\star \circ h^\star$ can be equivalently written as $f_s \circ h_s$ for some $f_s \in \mathcal{F}_s$.

**Theorem 1** (Weak-to-Strong Generalization under Realizability). *Let $h^\star : \mathbb{R}^d \to \mathbb{R}^{d^\star}$ be a ground truth representation map, and let $f^\star : \mathbb{R}^{d^\star} \to \mathbb{R}$ be a finetuning task of interest. Let $h_s : \mathbb{R}^d \to \mathbb{R}^{d_s}$ and $h_w : \mathbb{R}^d \to \mathbb{R}^{d_w}$ be the strong and weak model representation maps respectively. Given some data labeled by $f^\star \circ h^\star$, let $f_w \circ h_w$ be the function learnt by the weak model, for some arbitrary function $f_w : \mathbb{R}^{d_w} \to \mathbb{R}$. Now, for a convex set of functions $\mathcal{F}_s$ mapping $\mathbb{R}^{d_s}$ to $\mathbb{R}$ let*

$$f_{sw} = argmin_{f \in \mathcal{F}_s} \, d_\mathcal{P}(f \circ h_s, f_w \circ h_w) \tag{1}$$

*be the function learnt by the strong model under weak supervision. Lastly, let us assume that there exists $f_s \in \mathcal{F}_s$ such that $f_s \circ h_s = f^\star \circ h^\star$. Then, we have that*

$$d_\mathcal{P}(f_{sw} \circ h_s, f^\star \circ h^\star) \leq d_\mathcal{P}(f_w \circ h_w, f^\star \circ h^\star) - d_\mathcal{P}(f_{sw} \circ h_s, f_w \circ h_w). \tag{2}$$

On the left-hand side in (2) is the error of the weakly-supervised strong model on the true data. The first term on the right-hand side is the true error of the weak model, and the second term is the error of the weakly-supervised strong model on data labeled by the weak model (i.e., misfit). Thus, the inequality directly says that the weakly-supervised strong model improves over the weak model by (at least) an amount equal to the misfit. Note again that in practice, a popular way to finetune a pretrained model on task-specific data is by tuning the weights of only the last linear layer of the model. In these cases, $\mathcal{F}_s$ is simply the set of linear functions, which is convex. We emphasize that neither of $f^\star$ or $f_w$ need to belong to $\mathcal{F}_s$; as long as the strong model finds the *minimizer* over a convex set of the loss on the weakly labeled data (as in (1)), the inequality in (2) holds.

Next, we relax the realizability assumption that the target task $f^\star \circ h^\star$ belongs to the space of functions that the strong model optimizes over. Instead, suppose that by composing $h_s$ with functions in $\mathcal{F}_s$, it is possible for the strong model to get a small distance $\varepsilon$ to the target task. The strong model could obtain such a powerful representation map after having seen an abundance of pretraining data; the realizable case corresponds to $\varepsilon = 0$. We also relax the assumption that the strong model is able to obtain the true minimizer with respect to the data distribution $\mathcal{P}$ as in (1). In reality, we can imagine that the strong model only sees a finite sample labeled by the weak model, and obtains $\hat{f}_{sw}$ by minimizing the loss over this finite sample. Even with these relaxations, we can show that the same qualitative result as in Theorem 1 continues to hold, upto small error terms.

**Theorem 2** (Weak-to-Strong Generalization under Non-Realizability and Finite Samples). *Let $h^\star : \mathbb{R}^d \to \mathbb{R}^{d^\star}$ be a ground truth representation map, and let $f^\star : \mathbb{R}^{d^\star} \to \mathbb{R}$ be a finetuning task of interest. Let $h_s : \mathbb{R}^d \to \mathbb{R}^{d_s}$ and $h_w : \mathbb{R}^d \to \mathbb{R}^{d_w}$ be the strong and weak model representations respectively. Given some data labeled by $f^\star \circ h^\star$, let $f_w \circ h_w$ be the function learnt by the weak model, for some arbitrary function $f_w : \mathbb{R}^{d_w} \to \mathbb{R}$. For a convex set of functions $\mathcal{F}_s$ mapping $\mathbb{R}^{d_s} \to \mathbb{R}$, let*

$$f_s = argmin_{f \in \mathcal{F}_s} \, d_\mathcal{P}(f \circ h_s, f^\star \circ h^\star), \tag{3}$$

*and suppose that $d_\mathcal{P}(f_s \circ h_s, f^\star \circ h^\star) = \varepsilon$. Now, suppose we obtain $n$ weakly-labeled i.i.d. samples $(x_1, y_1), \ldots, (x_n, y_n)$, where each $x_i \sim \mathcal{P}$ and $y_i = f_w \circ h_w(x_i)$. Let*

$$\hat{f}_{sw} = argmin_{f \in \mathcal{F}_s} \frac{1}{n} \sum_{i=1}^{n} (f \circ h_s(x_i) - y_i)^2. \tag{4}$$

*Finally, assume that the range of $f^\star$, $f_w$ and all the functions in $\mathcal{F}_s$ is absolutely bounded. Then, we have that with probability at least $1 - \delta$ over the draw of $(x_1, y_1), \ldots, (x_n, y_n)$,*

$$d_\mathcal{P}(\hat{f}_{sw} \circ h_s, f^\star \circ h^\star) \leq d_\mathcal{P}(f_w \circ h_w, f^\star \circ h^\star) - d_\mathcal{P}(\hat{f}_{sw} \circ h_s, f_w \circ h_w)$$
$$+ O(\sqrt{\varepsilon}) + O\left(\frac{\mathcal{C}_{\mathcal{F}_s}}{n}\right)^{\frac{1}{4}} + O\left(\frac{\log(1/\delta)}{n}\right)^{\frac{1}{4}}, \tag{5}$$

where $\mathcal{C}_{\mathcal{F}_s}$ is a constant capturing the complexity of the function class $\mathcal{F}_s$, and the asymptotic notation is with respect to $\varepsilon \to 0, n \to \infty$.

As compared to (2), the bound in (5) has two sources of error terms: the first error term of $O(\sqrt{\varepsilon})$ arises (via standard triangle inequality arguments) due to the non-realizability assumption, and goes to zero as the strong model becomes stronger and more expressive. The latter two error terms arise (via standard uniform convergence arguments as in [TJJ20]) because the strong model only sees a finite weakly-labeled sample—these terms vanish too as the sample size becomes large.

## 4 Main Proof Technique

In this section, we outline the proof of realizable weak-to-strong generalization (Theorem 1). The proof of Theorem 2 uses the same main idea and is given in Appendix A. Recall that the strong model learns from a convex set $\mathcal{F}_s : \mathbb{R}^{d_s} \to \mathbb{R}$ of finetuning tasks. Recall also that we denote the strong model representation map by $h_s : \mathbb{R}^d \to \mathbb{R}^{d_s}$. Let $V_s = \{f \circ h_s : f \in \mathcal{F}_s\}$ be the set of all tasks in $\mathcal{F}_s$ composed with the strong model representation. We first observe that $V_s$ is also a convex set.

**Claim 3.** $V_s$ is a convex set.

*Proof.* Fix $f, g \in \mathcal{F}_s$, and consider $f \circ h_s, g \circ h_s \in V_s$. Fix any $\lambda \in [0, 1]$. Since $\mathcal{F}_s$ is a convex set, there exists $p \in \mathcal{F}_s$ such that for all $y \in \mathbb{R}^{d_s}$, $p(y) = \lambda f(y) + (1 - \lambda)g(y)$. Now, fix any $x \in \mathbb{R}^d$. Then, we have that

$$\lambda(f \circ h_s)(x) + (1 - \lambda)(g \circ h_s)(x) = \lambda f(h_s(x)) + (1 - \lambda)g(h_s(x)) = p(h_s(x)) = (p \circ h_s)(x),$$

and hence $\lambda(f \circ h_s) + (1 - \lambda)(g \circ h_s) = p \circ h_s \in V_s$. $\qquad\square$

We are then ready to prove Theorem 1.

*Proof of Theorem 1.* The setting under consideration is depicted in Figure 1. Since we assume realizability, $f^\star \circ h^\star \in V_s$. Let $A = d_{\mathcal{P}}(f_{sw} \circ h_s, f^\star \circ h^\star)$, $B = d_{\mathcal{P}}(f_{sw} \circ h_s, f_w \circ h_w)$ and $C = d_{\mathcal{P}}(f_w \circ h_w, f^\star \circ h^\star)$. We want to show that $C \geq A + B$. Recall that

$$f_{sw} = \operatorname{argmin}_{f \in \mathcal{F}_s} d_{\mathcal{P}}(f \circ h_s, f_w \circ h_w).$$

In other words, $f_{sw} \circ h_s$ is the *projection* of $f_w \circ h_w$ onto the convex set $V_s$. We can therefore apply the "Pythagorean theorem" for projections onto a convex set [Haz16, Theorem 2.1].

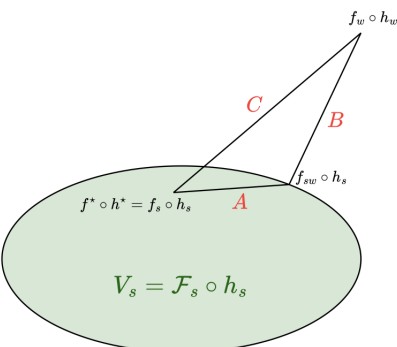

Figure 1: $f_{sw} \circ h_s$ is the projection of $f_w \circ h_w$ onto the convex set $V_s$.

Concretely, for any $g \in V_s$, observe that

$$
\begin{aligned}
d_{\mathcal{P}}(f_w \circ h_w, g) &= \mathbb{E}_{x \sim \mathcal{P}}(g(x) - (f_w \circ h_w)(x))^2 \\
&= \mathbb{E}_{x \sim \mathcal{P}}(g(x) - (f_{sw} \circ h_s)(x) + (f_{sw} \circ h_s)(x) - (f_w \circ h_w)(x))^2 \\
&= \mathbb{E}_{x \sim \mathcal{P}}(g(x) - (f_{sw} \circ h_s)(x))^2 + \mathbb{E}_{x \sim \mathcal{P}}((f_{sw} \circ h_s)(x) - (f_w \circ h_w)(x))^2 \\
&\quad + 2 \cdot \mathbb{E}_{x \sim \mathcal{P}}\left[(g(x) - (f_{sw} \circ h_s)(x))((f_{sw} \circ h_s)(x) - (f_w \circ h_w)(x))\right] \\
&= d_{\mathcal{P}}(f_{sw} \circ h_s, g) + d_{\mathcal{P}}(f_{sw} \circ h_s, f_w \circ h_w) \\
&\quad + 2 \cdot \mathbb{E}_{x \sim \mathcal{P}}\left[(g(x) - (f_{sw} \circ h_s)(x))((f_{sw} \circ h_s)(x) - (f_w \circ h_w)(x))\right]. \quad (6)
\end{aligned}
$$

But note also that by definition of projection, $d_{\mathcal{P}}(f_w \circ h_w, g) \geq d_{\mathcal{P}}(f_{sw} \circ h_s, f_w \circ h_w)$, and hence

$$d_{\mathcal{P}}(f_{sw} \circ h_s, g) + 2 \cdot \mathbb{E}_{x \sim \mathcal{P}}\left[(g(x) - (f_{sw} \circ h_s)(x))((f_{sw} \circ h_s)(x) - (f_w \circ h_w)(x))\right] \geq 0. \quad (7)$$

Now, fix $t \in (0,1)$, and consider the function

$$w(t) = f_{sw} \circ h_s + t \cdot (f^{\star} \circ h^{\star} - f_{sw} \circ h_s).$$

Namely, for any $x$, $w(t)(x) = (f_{sw} \circ h_s)(x) + t \cdot ((f^{\star} \circ h^{\star})(x) - (f_{sw} \circ h_s)(x))$. Because $V_s$ is a convex set (Claim 3), $w(t) \in V_s$. Also,

$$d_{\mathcal{P}}(f_{sw} \circ h_s, w(t)) = \mathbb{E}_{x \sim \mathcal{P}}((f_{sw} \circ h_s)(x) - w(t)(x))^2$$
$$= t^2 \cdot \mathbb{E}_{x \sim \mathcal{P}}((f^{\star} \circ h^{\star})(x) - (f_{sw} \circ h_s)(x))^2.$$

Hence, substituting $w(t)$ for $g$ in (7), we get

$$t^2 \cdot \mathbb{E}_{x \sim \mathcal{P}}((f^{\star} \circ h^{\star})(x) - (f_{sw} \circ h_s)(x))^2$$
$$+ 2t \cdot \mathbb{E}_{x \sim \mathcal{P}}\left[((f^{\star} \circ h^{\star})(x) - (f_{sw} \circ h_s)(x))((f_{sw} \circ h_s)(x) - (f_w \circ h_w)(x))\right] \geq 0.$$

Taking the limit as $t \downarrow 0$, we get that

$$\mathbb{E}_{x \sim \mathcal{P}}\left[((f^{\star} \circ h^{\star})(x) - (f_{sw} \circ h_s)(x))((f_{sw} \circ h_s)(x) - (f_w \circ h_w)(x))\right] \geq 0 \quad (8)$$

Substituting $f^{\star} \circ h^{\star}$ for $g$ in (6), and using (8), we obtain the desired result

$$d_{\mathcal{P}}(f_w \circ h_w, f^{\star} \circ h^{\star}) \geq d_{\mathcal{P}}(f_{sw} \circ h_s, f^{\star} \circ h^{\star}) + d_{\mathcal{P}}(f_{sw} \circ h_s, f_w \circ h_w).$$

$\square$

## 5   Experiments

We perform experiments[4] on synthetically generated data as well as real-world molecular prediction and natural language datasets to verify the guarantees on weak-to-strong generalization given by our theorems. The results for the natural language tasks are given in Appendix C.

### 5.1   Synthetic Experiments

We set the target data representation $h^{\star} : \mathbb{R}^8 \to \mathbb{R}^{16}$ to be a randomly initialized 5-layer multi-layer perceptron (MLP) with ReLU activations, with input dimension 8 and hidden layer dimension 16. The class $\mathcal{F}_s$ of finetuning tasks from which the strong model (as well as the weak model) learns is simply the class of linear functions from $\mathbb{R}^{16} \to \mathbb{R}$; $\mathcal{F}_s$ is thus a convex set (see Appendix D for instances where $\mathcal{F}_s$ is a non-convex set). The marginal data distribution $\mathcal{P}$ in our experiments is always $\mathcal{N}(0, \sigma^2 I)$. To ensure that the data is well-spread, we set $\sigma = 500$.

**Representation Learning.** We experiment with two different ways of obtaining the weak and strong representations $h_w$ and $h_s$:

(1) **Pretraining:** We randomly sample $T$ finetuning tasks $f^{(1)}, \ldots, f^{(T)} \in \mathcal{F}_s$. For each $t \in [T]$, we generate data $\{x_j^{(t)}, y_j^{(t)}\}_{j=1}^{N_r}$, where $x_j^{(t)} \sim \mathcal{P}$ and $y_j^{(t)} = f^{(t)} \circ h^{\star}(x_j^{(t)})$. Loosely following [TJJ20], we obtain $h_w$ and $h_s$ as

$$h_k = \mathrm{argmin}_{h \in \mathcal{H}_k} \frac{1}{T \cdot N_r} \sum_{t=1}^{T} \sum_{j=1}^{N_r} (f^{(t)} \circ h(x_j^{(t)}) - y_j^{(t)})^2 \qquad \text{for } k \in \{w, s\}. \quad (9)$$

We set $\mathcal{H}_w$ and $\mathcal{H}_s$ (both $\mathbb{R}^8 \to \mathbb{R}^{16}$) to be the classes of 2-layer and 8-layer neural networks respectively with ReLU activations and hidden dimension 16. We obtain $h_w$ and $h_s$ via gradient descent on the representation parameters to find the minimizers in (9). We set $T = 10, N_r = 2000$. Additionally, we also consider the *realizable* setting (Theorem 1), where we explicitly set $h_s = h^{\star}$, and only obtain $h_w$ as above.

(2) **Perturbations:** We also consider another way to obtain the weak and strong representations as direct perturbations of $h^{\star}$. Namely, we perturb every parameter in every weight matrix in $h^{\star}$ by independent Gaussian noise $\mathcal{N}(0, \sigma_s^2)$ to obtain $h_s$. Similarly, we obtain $h_w$ by perturbing each parameter in $h^{\star}$ by $\mathcal{N}(0, \sigma_w^2)$. Ideally, we want the strong representation $h_s$ to be a closer approximation of $h^{\star}$ than $h_w$. Hence, we set $\sigma_s = 0.01$ and $\sigma_w = 0.05$.

---

[4] https://github.com/chogba/wtsg-regression

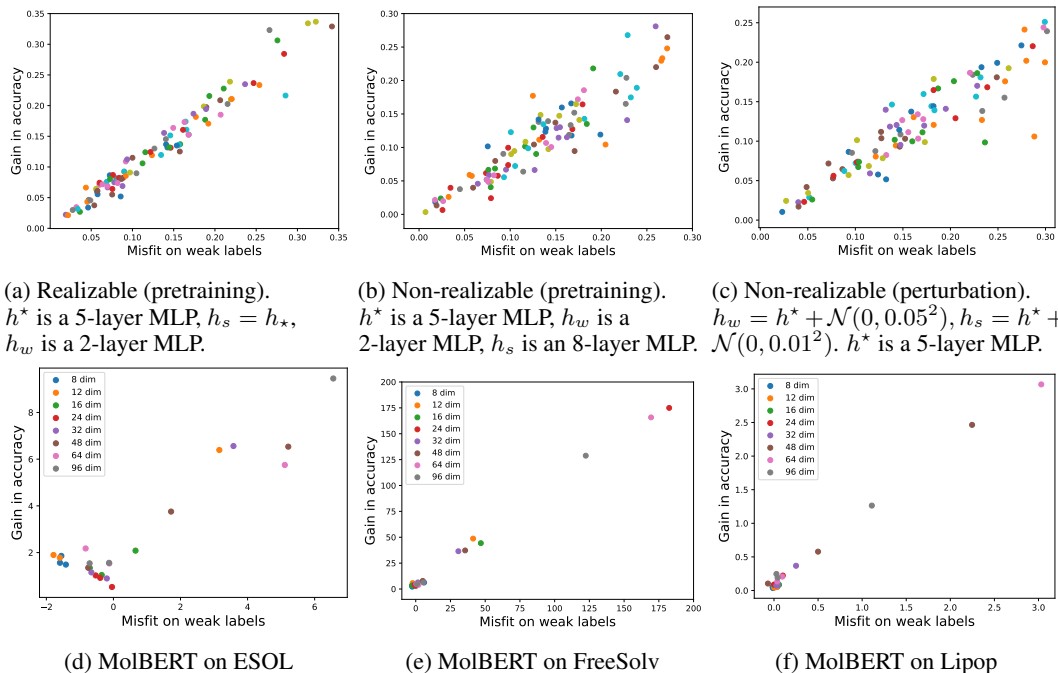

Figure 2: (a),(b),(c) Experiments on synthetic data. (d),(e),(f) QSAR tasks over MolBERT representations on the ESOL, FreeSolv and Lipop datasets. For each dataset, ChemBench [Wan20] provides three different train, test and validation splits; multiple points of the same color correspond to weak-to-strong supervision for the same weak model (as specified in legend) across these splits.

**Weak Model Finetuning.** Once the representations $h_w$ and $h_s$ have been obtained and fixed, we randomly generate $M$ *new* finetuning tasks $f^{(1)}, \ldots, f^{(M)} \in \mathcal{F}_s$, and obtain data $\{x_j^{(i)}, y_j^{(i)}\}_{j=1}^{N_f}$ for each of these tasks. Here again, $x_j^{(i)} \sim \mathcal{P}$ and $y_j^{(i)} = f^{(i)} \circ h^\star(x_j^{(i)})$. We set $M = 100, N_f = 2000$. For each task, we train the weak model on the data generated for the task, to obtain

$$f_w^{(i)} = \text{argmin}_{f \in \mathcal{F}_s} \frac{1}{N_f} \sum_{j=1}^{N_f} (f \circ h_w(x_j^{(i)}) - y_j^{(i)})^2. \tag{10}$$

Here, the representation parameters $h_w$ are frozen, and $f_w^{(i)}$ is obtained via gradient descent. Note again that we are training the weak models on *true* data labeled by the finetuning task $f^{(i)} \circ h^\star$.

**Weak-to-Strong Supervision.** Once our weak models are trained for each finetuning task, we generate *weakly labeled data*. That is, for each $i \in [M]$, we generate $\{\tilde{x}_j^{(i)}, \tilde{y}_j^{(i)}\}_{j=1}^{N_f}$ where $\tilde{x}_j^{(i)} \sim \mathcal{P}$. But crucially, $\tilde{y}_j^{(i)} = f_w^{(i)} \circ h_w(\tilde{x}_j^{(i)})$. We now train our strong models on this weakly labeled data. Namely, keeping the strong representation $h_s$ fixed, we obtain, via gradient descent again

$$f_{sw}^{(i)} = \text{argmin}_{f \in \mathcal{F}_s} \frac{1}{N_f} \sum_{j=1}^{N_f} (f \circ h_s(\tilde{x}_j^{(i)}) - \tilde{y}_j^{(i)})^2. \tag{11}$$

At this point, our weak-to-strong training procedure is complete.

**Evaluation.** For each finetuning task, we wish to evaluate the accuracy of our weak-to-strong model $f_{sw}^{(i)} \circ h_s$ with respect to the true task $f^{(i)} \circ h^\star$.

Towards this, we estimate 3 quantities:

    (a) Error of the weak-to-strong model $f_{sw}^{(i)} \circ h_s$ on the *true* finetuning task: $\mathbb{E}_{x \sim \mathcal{P}}(f_{sw}^{(i)} \circ h_s(x) - f^{(i)} \circ h^\star(x))^2$.

(b) Error of the weak model $f_w^{(i)} \circ h_w$ on the *true* finetuning task: $\mathbb{E}_{x \sim \mathcal{P}}(f_w^{(i)} \circ h_w(x) - f^{(i)} \circ h^\star(x))^2$.

(c) Misfit error of the weak-to-strong model on the weakly labeled data: $\mathbb{E}_{x \sim \mathcal{P}}(f_{sw}^{(i)} \circ h_s(x) - f_w^{(i)} \circ h_w(x))^2$.

Each of these quantities are estimated from a fresh sample of size $N_f$ drawn from $\mathcal{P}$. For each task $i \in [M]$, we plot the difference (b)-(a), namely the **Gain in Accuracy**, on the y-axis, versus the **Misfit** (c) on the x-axis. Figure 2a has the results for the realizable case where $h_s = h^\star$ and $h_w$ is obtained by pretraining. Figure 2b has the results for the non-realizable case where both $h_w$ and $h_s$ are obtained by pretraining. Figure 2c has the results for the non-realizable case where $h_w$ and $h_s$ are obtained by directly perturbing the weights in $h^\star$. For reference, recall that Theorem 2 indicates that the gain in accuracy is (upto error terms) *at least* the misfit. The plots in Figure 2 suggest that the gain is more or less *exactly* the misfit, which is in agreement with our theory!

## 5.2 Molecular Prediction

We also validate our conceptual insights on real-world molecular prediction datasets. Specifically, we follow the Quantitative Structure-Activity Relationship (QSAR) task setup in the MolBERT [FEG+20] paper. These tasks involve predicting physical properties of molecules like solubility, lipophilicity, etc. We consider three regression datasets: ESOL, FreeSolv and Lipop. These datasets are part of the MoleculeNet [WRF+18] benchmark suite, and have been curated into train, test and validation splits by ChemBench [Wan20]. The MolBERT paper provides weights for a standard-size BERT [DCLT18] architecture (hidden dimension 768, 12 layers, 12 attention heads) pretrained for 100 epochs on the GuacaMol [BFSV19] dataset. We use these weights as the strong representation $h_s$. For the weak representations $h_w$, we run the pretraining pipeline for substantially smaller transformer architectures and lesser compute time. Specifically, we consider transformers with just 2 layers and 2 attention heads, and vary the hidden size in $\{8, 12, 16, 32, 48, 64, 96\}$. For each of these settings, we run the pretraining tasks for a mere 2 epochs to obtain different weak representations $h_w$.

Once we have the representations $h_s$ and $h_w$, we can finetune a linear layer on top of these for each of the three regression datasets. We run the entire weak-to-strong supervision pipeline from above, where we weakly supervise the strong model $h_s$ on labels given by each of the weak models $h_w$. The results are given in Figures 2d, 2e and 2f. Again, we see that the gain in accuracy of the weakly supervised strong models is accurately characterized by their misfit on the weak labels.

We were also able to see an otherwise useful algorithmic insight in these experiments. Consider a setting where we have at our disposal various weak models, and have performed the weak-to-strong supervision pipeline separately on each of them. We now want to deploy one of the weakly trained strong models; our goal is to choose the one that gets the least error on the true data distribution. Recall that Theorem 2 guarantees that the error of a weakly supervised strong model is upper bounded (upto error terms) by the difference between the weak model's error and misfit. This suggests a natural heuristic: sort the strong models by the difference between the corresponding weak model's error and the misfit, and choose the one for which this quantity is smallest. We observed that this heuristic ends up working quite well—Table 1 shows the numbers for the Lipop dataset, while the results for ESOL and FreeSolv are in Appendix B.

## 5.3 Strong-to-Weak Generalization and Low Sample Regime

In our simulations, we also consider an additional thought experiment, where we reverse the weak and strong models. That is, in the non-realizable case with pretraining (Figure 2b), we can have $\mathcal{H}_w$ be the class of 8-layer MLPs, and $\mathcal{H}_s$ be the class of 2-layer MLPs. Similarly, in the case with perturbations (Figure 2c), we can set $\sigma_w = 0.01$, and $\sigma_s = 0.05$. In this case, because the weak models have now become powerful, and can represent the true data well, the weak labels are essentially the *true* labels. Hence, if we were to obtain the same plots, we would now expect the misfit on weak labels to essentially correspond to the *loss* in accuracy of the strong model on true data, compared to the weak model. This is confirmed in Figures 3a and 3b: the plots are mirror reflections of Figures 2b and 2c!

Now, suppose that we are in a setting where the number of samples available for the representation learning task is scarce. Concretely, consider the original setting of Figure 2b with $\mathcal{H}_w$ and $\mathcal{H}_s$ back to being 2-layer and 8-layer MLPs respectively. Recall that for learning the representations $h_w, h_s$,

| Hidden dimension | Weak error - Misfit | True error of weakly-supervised strong model |
|---|---|---|
| 96 | **$0.8969 \pm 0.0327$** | **$1.0713 \pm 0.0489$** |
| 48 | $0.9731 \pm 0.0707$ | $1.1293 \pm 0.0418$ |
| 24 | $1.0331 \pm 0.0449$ | $1.1204 \pm 0.0261$ |
| 64 | $1.0619 \pm 0.0441$ | $1.1436 \pm 0.0124$ |
| 32 | $1.0624 \pm 0.0527$ | $1.1302 \pm 0.0220$ |
| 16 | $1.1456 \pm 0.0276$ | $1.1950 \pm 0.0484$ |
| 12 | $1.1499 \pm 0.0177$ | $1.1869 \pm 0.0297$ |
| 8 | $1.1958 \pm 0.0194$ | $1.2396 \pm 0.0310$ |

Table 1: Heuristic rule to choose among different weakly-supervised models finetuned on Lipop: choose the strong model that has the smallest difference (averaged across the 3 splits) between weak model error and misfit ($\pm$ is the std across splits). As we see, this model has the smallest true error.

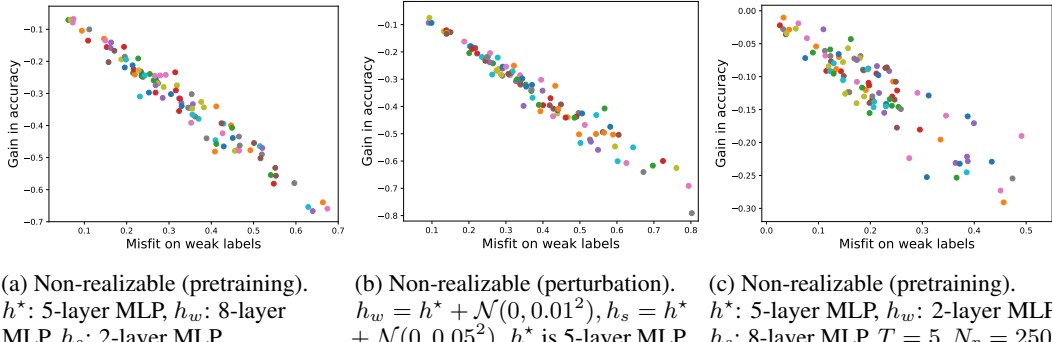

(a) Non-realizable (pretraining). $h^\star$: 5-layer MLP, $h_w$: 8-layer MLP, $h_s$: 2-layer MLP.

(b) Non-realizable (perturbation). $h_w = h^\star + \mathcal{N}(0, 0.01^2), h_s = h^\star + \mathcal{N}(0, 0.05^2)$. $h^\star$ is 5-layer MLP.

(c) Non-realizable (pretraining). $h^\star$: 5-layer MLP, $h_w$: 2-layer MLP, $h_s$: 8-layer MLP. $T = 5, N_r = 250$.

Figure 3: Strong-to-weak generalization. The roles of the weak and strong models have reversed.

we sampled $T = 10$ finetuning tasks $f^{(1)}, \ldots, f^{(T)} \in \mathcal{F}_s$, and obtained $N_r = 2000$ samples labeled according to each $f^{(t)} \circ h^\star$. Now instead, consider setting $T = 5, N_r = 250$. The number of samples $N_f$ in the weak model finetuning and weak-to-strong supervision stages is still maintained at $N_f = 2000$. We run the entire weak-to-strong supervision pipeline for this parameter setting. The rationale is that, when the representation learning task is data-deprived, the weak model, by virtue of being simpler, learns a better representation than the strong model, which is more complex. Indeed, this is what we observed, as shown in Figure 3c. Observe that the trend in the plot is very similar to Figures 3a and 3b, where we had *explicitly* swapped the weak and strong models. This suggests that in the low-sample regime too, the weak and strong models have reversed roles. Thus, the definition of weak and strong models in the framework of weak-to-strong generalization should not solely be based on expressive power; instead, these roles should be assigned based on the quality of representations.

## 6    Conclusion

Employing a representation-theoretic perspective, we characterized the gain in performance in weak-to-strong generalization. Our results apply in the setting of learning real-valued functions with the least squares loss, where the strong model learns the finetuning task by optimizing over a convex set of functions. We quantify the gain in accuracy of the weakly-supervised strong model over its weak supervisor in terms of the misfit between the strong and weak models.

Our work has natural limitations. Our theorems notably do not apply when the set from which the strong model learns the finetuning task is not convex. Nevertheless, our experiments in Appendix D do suggest that our results should (at least qualitatively) hold even beyond the convex case. Our work also does not address classification tasks (see also Appendix E), and it would be interesting to see if similar results could be obtained for more general loss functions. Finally, while we do demonstrate results on real-world datasets, we anticipate that significantly larger-scale experiments on regression datasets used to train modern AI models will yield further interesting insights.

## Acknowledgments and Disclosure of Funding

This work is supported by Moses Charikar and Gregory Valiant's Simons Investigator Awards. The authors would like to thank Percy Liang and Tengyu Ma for helpful discussions, and also the anonymous reviewers for engaging with the paper and pointing out references.

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

## A    Non-realizable Weak-to-Strong Generalization

*Proof of Theorem 2.*  The setting under consideration is depicted in Figure 4.

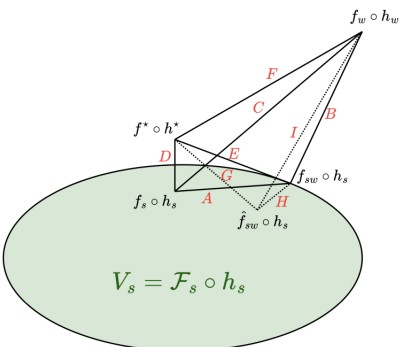

Figure 4: Non-realizable weak-to-strong generalization where $f^\star \circ h^\star \notin V_s$, and we use a finite sample to perform weak-to-strong supervision.  The Pythagorean theorem, along with uniform convergence and triangle inequalities, yield the desired result.

Let

$$A = d_\mathcal{P}(f_{sw} \circ h_s, f_s \circ h_s)$$
$$B = d_\mathcal{P}(f_{sw} \circ h_s, f_w \circ h_w)$$
$$C = d_\mathcal{P}(f_w \circ h_w, f_s \circ h_s)$$
$$D = d_\mathcal{P}(f_s \circ h_s, f^\star \circ h^\star) = \varepsilon$$
$$E = d_\mathcal{P}(f_{sw} \circ h_s, f^\star \circ h^\star)$$
$$F = d_\mathcal{P}(f_w \circ h_w, f^\star \circ h^\star)$$
$$G = d_\mathcal{P}(\hat{f}_{sw} \circ h_s, f^\star \circ h^\star)$$
$$H = d_\mathcal{P}(\hat{f}_{sw} \circ h_s, f_{sw} \circ h_s)$$
$$I = d_\mathcal{P}(\hat{f}_{sw} \circ h_s, f_w \circ h_w).$$

Note that by virtue of the range of $f^\star, f_w$ and all functions in $\mathcal{F}_s$ being absolutely bounded, $d_\mathcal{P}$ is also bounded above by a constant.

We want to show that $G \leq F - I + O(\sqrt{\varepsilon}) + O\left(\frac{c_{\mathcal{F}_s}}{n}\right)^{\frac{1}{4}}$.

From the proof of Theorem 1, we know that

$$C \geq A + B. \tag{12}$$

But note also that by a "triangle inequality" (Claim 5),

$$\sqrt{E} \leq \sqrt{A} + \sqrt{D}$$
$$\implies \quad E \leq A + D + 2\sqrt{AD}. \tag{13}$$

Combining (12) and (13), we get

$$E \leq C + D - B + 2\sqrt{AD}. \tag{14}$$

By the same triangle inequality argument,

$$\sqrt{C} \leq \sqrt{D} + \sqrt{F} \tag{15}$$
$$\implies \quad C \leq D + F + 2\sqrt{DF}. \tag{16}$$

Substituting (16) in (14), we get

$$E \leq F - B + 2D + 2\sqrt{DF} + 2\sqrt{AD} \tag{17}$$

By a uniform convergence argument (Lemma 4), we have that with probability at least $1 - \delta$ over the draw of $\{(x_1, y_1), \ldots, (x_n, y_n)\}$ that were used to construct $\hat{f}_{sw}$,

$$I \leq B + O\left(\sqrt{\frac{\mathcal{C}_{\mathcal{F}_s}}{n}}\right) + O\left(\sqrt{\frac{\log(1/\delta)}{n}}\right). \tag{18}$$

Substituting (18) in (17), we get

$$E \leq F - I + 2D + 2\sqrt{DF} + 2\sqrt{AD} + O\left(\sqrt{\frac{\mathcal{C}_{\mathcal{F}_s}}{n}}\right) + O\left(\sqrt{\frac{\log(1/\delta)}{n}}\right). \tag{19}$$

Because $f_{sw} \circ h_s$ is the projection of $f_w \circ h_w$ onto $V_s$, we know (e.g., by the argument in the proof of Theorem 1) that

$$I \geq H + B. \tag{20}$$

Combining (18) and (20), we get

$$H \leq O\left(\sqrt{\frac{\mathcal{C}_{\mathcal{F}_s}}{n}}\right) + O\left(\sqrt{\frac{\log(1/\delta)}{n}}\right). \tag{21}$$

Finally, by another triangle inequality, note that

$$\sqrt{G} \leq \sqrt{E} + \sqrt{H}$$
$$\implies \quad G \leq E + H + 2\sqrt{EH}. \tag{22}$$

Substituting (19) in (22), we get

$$G \leq F - I + 2D + 2\sqrt{DF} + 2\sqrt{AD} + O\left(\sqrt{\frac{\mathcal{C}_{\mathcal{F}_s}}{n}}\right) + O\left(\sqrt{\frac{\log(1/\delta)}{n}}\right) + H + 2\sqrt{EH}$$

$$\leq F - I + 2D + 2\sqrt{DF} + 2\sqrt{AD} + 2\sqrt{E} \cdot \left[O\left(\frac{\mathcal{C}_{\mathcal{F}_s}}{n}\right)^{\frac{1}{4}} + O\left(\frac{\log(1/\delta)}{n}\right)^{\frac{1}{4}}\right] \quad \text{(from (21))}$$

$$\leq F - I + O(\sqrt{\varepsilon}) + O\left(\frac{\mathcal{C}_{\mathcal{F}_s}}{n}\right)^{\frac{1}{4}} + O\left(\frac{\log(1/\delta)}{n}\right)^{\frac{1}{4}},$$

where in the last inequality, we substituted $D = \varepsilon$, used that $d_{\mathcal{P}}$ is bounded above by a constant, and instantiated asymptotics with respect to $\varepsilon \to 0$ and $n \to \infty$. $\quad\square$

**Lemma 4** (Uniform Convergence). *Let $(x_1, y_1), \ldots, (x_n, y_n)$ be an i.i.d. training sample, where each $x_i \sim \mathcal{P}$ and $y_i = g(x_i)$ for some unknown target function $g$. For a fixed strong model representation $h_s$, let*

$$f_{sw} = \operatorname{argmin}_{f \in \mathcal{F}_s} d_{\mathcal{P}}(f \circ h_s, g), \qquad \hat{f}_{sw} = \operatorname{argmin}_{f \in \mathcal{F}_s} \frac{1}{n}(f \circ h_s(x_i) - y_i)^2.$$

*Assume that the range of $g$ and functions in $\mathcal{F}_s$ is absolutely bounded. Then, with probability at least $1 - \delta$ over the draw of $(x_1, y_1), \ldots, (x_n, y_n)$, we have that*

$$\left| d_{\mathcal{P}}(\hat{f}_{sw} \circ h_s, g) - d_{\mathcal{P}}(f_{sw} \circ h_s, g) \right| \leq O\left(\sqrt{\frac{\mathcal{C}_{\mathcal{F}_s}}{n}}\right) + O\left(\sqrt{\frac{\log(1/\delta)}{n}}\right).$$

*Proof.* Given sample $(x_1, y_1), \ldots, (x_n, y_n)$ where each $x_i \sim \mathcal{P}$ and $y_i = g(x_i)$, define

$$\hat{d}_{\mathcal{P}}(s, g) = \frac{1}{n} \sum_{i=1}^{n} (s(x_i) - y_i)^2$$

for any function $s$. Then, we have that

$$d_{\mathcal{P}}(\hat{f}_{sw} \circ h_s, g) - d_{\mathcal{P}}(f_{sw} \circ h_s, g) = \underbrace{d_{\mathcal{P}}(\hat{f}_{sw} \circ h_s, g) - \hat{d}_{\mathcal{P}}(\hat{f}_{sw} \circ h_s, g)}_{a}$$

$$+ \underbrace{\hat{d}_{\mathcal{P}}(\hat{f}_{sw} \circ h_s, g) - \hat{d}_{\mathcal{P}}(f_{sw} \circ h_s, g)}_{b}$$

$$+ \underbrace{\hat{d}_{\mathcal{P}}(f_{sw} \circ h_s, g) - d_{\mathcal{P}}(f_{sw} \circ h_s, g)}_{c}. \qquad (23)$$

By the definition of $\hat{f}_{sw}$, the second term $b \leq 0$ in (23). The terms $a$ and $c$ measure the difference between the empirical distance and true population distance, and can be controlled by a standard uniform convergence argument, as in [TJJ20, Theorem 2]. For completeness, we sketch the main parts of the argument here.

Let $S = \{(x_1, y_1), \ldots, (x_n, y_n)\}$, where $x_i \sim \mathcal{P}$ and $y_i = g(x_i)$. Using McDiarmid's inequality, and by a double-sampling and symmetrization argument, it first holds that with probability at least $1 - \delta$,

$$\sup_{f \in \mathcal{F}_s} |\hat{d}_{\mathcal{P}}(f \circ h_s, g) - d_{\mathcal{P}}(f \circ h_s, g)| \leq O\left(\mathcal{R}_n(l(\mathcal{F}_s \circ h_s))\right) + O\left(\sqrt{\frac{\log(1/\delta)}{n}}\right),$$

where $\mathcal{R}_n(l(\mathcal{F}_s \circ h_s))$ is the *Rademacher complexity* of the loss class of $\mathcal{F}_s \circ h_s$:

$$\mathcal{R}_n(l(\mathcal{F}_s \circ h_s)) = \mathbb{E}_S \mathbb{E}_{\varepsilon_i \sim \{-1,1\}} \sup_{f \in \mathcal{F}_s} \frac{1}{n} \sum_{i=1}^n \varepsilon_i \cdot (f \circ h_s(x_i) - y_i)^2.$$

We can then use the assumption that the range of $g$ and $\mathcal{F}_s$ is absolutely bounded, which implies that the squared loss function is both bounded and Lipschitz in both arguments. This allows us to use the contraction principle [LT13, Theorem 4.12] so as to move from the Rademacher complexity of the loss class $l(\mathcal{F}_s \circ h_s)$ to that of $\mathcal{F}_s \circ h_s$ itself, and claim that with probability at least $1 - \delta$,

$$\sup_{f \in \mathcal{F}_s} |\hat{d}_{\mathcal{P}}(f \circ h_s, g) - d_{\mathcal{P}}(f \circ h_s, g)| \leq O\left(\mathcal{R}_n(\mathcal{F}_s \circ h_s)\right) + O\left(\sqrt{\frac{\log(1/\delta)}{n}}\right) \qquad (24)$$

Finally, the Rademacher complexity $\mathcal{R}_n(\mathcal{F}_s \circ h_s)$ can be upper bounded by a quantity known as the *worst-case Gaussian complexity* of $\mathcal{F}_s$; in any case, for a majority of parametric function classes $\mathcal{F}_s$, this quantity scales as $\sqrt{\frac{C_{\mathcal{F}_s}}{n}}$, where $C_{\mathcal{F}_s}$ is a constant capturing the inherent complexity of $\mathcal{F}_s$. Plugging this into (24) yields the desired bound. $\qquad \square$

**Claim 5** ("Triangle Inequality"). *For any functions $f, g, h$,*

$$\sqrt{d_{\mathcal{P}}(f, g)} \leq \sqrt{d_{\mathcal{P}}(f, h)} + \sqrt{d_{\mathcal{P}}(h, g)}.$$

*Proof.*

$$\left(\sqrt{d_{\mathcal{P}}(f, h)} + \sqrt{d_{\mathcal{P}}(h, g)}\right)^2 = \mathbb{E}_{x \sim \mathcal{P}}(f(x) - h(x))^2 + \mathbb{E}_{x \sim \mathcal{P}}(h(x) - g(x))^2$$

$$+ 2\sqrt{\mathbb{E}_{x \sim \mathcal{P}}(f(x) - h(x))^2 \mathbb{E}_{x \sim \mathcal{P}}(h(x) - g(x))^2}, \qquad (25)$$

but also,

$$d_{\mathcal{P}}(f, g) = \mathbb{E}_{x \sim \mathcal{P}}(f(x) - g(x))^2$$

$$= \mathbb{E}_{x \sim \mathcal{P}}(f(x) - h(x) + h(x) - g(x))^2$$

$$= \mathbb{E}_{x \sim \mathcal{P}}(f(x) - h(x))^2 + \mathbb{E}_{x \sim \mathcal{P}}(h(x) - g(x))^2 + 2 \cdot \mathbb{E}_{x \sim \mathcal{P}}[(f(x) - h(x))(h(x) - g(x))]$$

$$= \left(\sqrt{d_{\mathcal{P}}(f, h)} + \sqrt{d_{\mathcal{P}}(h, g)}\right)^2 + 2 \cdot \mathbb{E}_{x \sim \mathcal{P}}[(f(x) - h(x))(h(x) - g(x))]$$

$$- 2\sqrt{\mathbb{E}_{x \sim \mathcal{P}}(f(x) - h(x))^2 \mathbb{E}_{x \sim \mathcal{P}}(h(x) - g(x))^2} \quad \text{(using (25))}$$

$$\leq \left(\sqrt{d_{\mathcal{P}}(f, h)} + \sqrt{d_{\mathcal{P}}(h, g)}\right)^2,$$

where the last step uses the Cauchy-Schwarz inequality. $\qquad \square$

## B    Heuristic to Choose Among Weak Models

Similar to Table 1, we report additional results on the ESOL and FreeSolv datasets about heuristically choosing amongst different weakly-supervised models based on the one that minimizes the upper bound given by Theorem 1 (upto error terms). The numbers are averaged across 3 train-test-validation splits. For ESOL (Table 2), weakly supervising with the weak model that results in the smallest difference between weak error and misfit also results in the strong model with least error. This was not exactly the case with FreeSolv (Table 3); however, we can still see the general trend that weak models that exhibit a smaller upper bound also tend to result in strong models that have better errors. We note that the upper bound in some cases is negative, which suggests that the error terms (as in Theorem 2) are non-negligible.

| Hidden dimension | Weak error - Misfit | True error of strong model trained on weak |
|:---:|:---:|:---:|
| 96 | $\mathbf{-0.8880 \pm 0.4314}$ | $\mathbf{1.3914 \pm 0.0906}$ |
| 48 | $-0.3634 \pm 0.1884$ | $1.4565 \pm 0.1860$ |
| 32 | $-0.3016 \pm 0.5698$ | $1.6583 \pm 0.2160$ |
| 64 | $-0.2450 \pm 0.7978$ | $1.5232 \pm 0.2291$ |
| 12 | $0.0083 \pm 0.2838$ | $3.4396 \pm 0.1433$ |
| 16 | $0.2582 \pm 0.1028$ | $1.8794 \pm 0.2185$ |
| 8 | $0.2618 \pm 0.2065$ | $3.4200 \pm 0.0753$ |
| 24 | $0.4904 \pm 0.1188$ | $1.6342 \pm 0.3086$ |

Table 2: ESOL

| Hidden dimension | Weak error - Misfit | True error of strong model trained on weak |
|:---:|:---:|:---:|
| 32 | $\mathbf{-0.1588 \pm 1.0254}$ | $4.9967 \pm 0.5656$ |
| 96 | $1.0896 \pm 1.0111$ | $4.7828 \pm 2.2787$ |
| 64 | $2.9062 \pm 2.8130$ | $\mathbf{3.8404 \pm 0.5906}$ |
| 12 | $2.9758 \pm 2.8664$ | $9.5102 \pm 2.4511$ |
| 48 | $3.0038 \pm 1.3063$ | $4.5212 \pm 0.3936$ |
| 16 | $4.8560 \pm 2.5910$ | $7.3637 \pm 1.1048$ |
| 24 | $6.4952 \pm 4.9463$ | $6.5863 \pm 1.2709$ |
| 8 | $6.7586 \pm 0.9997$ | $10.4474 \pm 1.6261$ |

Table 3: FreeSolv

## C    Experiments on NLP Datasets

We include additional experimental results on NLP regression tasks corresponding to two publicly available natural language datasets. The gain-misfit plots in these experiments largely follow the same trend as in Section 5.1 and Section 5.2.

### C.1    Essay Scoring

We use the dataset from the Kaggle competition "Feedback Prize - English Language Learning" [FMB$^+$22] conducted in 2022. Here, the task is to assign a score to essays written by 8th-12th grade English Language Learners (ELLs). An essay is assigned a score from 1-5 on each of 6 rubrics: cohesion, syntax, vocabulary, phraseology, grammar, conventions. Thus, a separate regression task can be obtained by treating each of these scores as the target label. The data source provides labels only for the train split; hence, we randomly do a 70-30 split of this data into train and test data.

We fix the strong model representation $h_s$ to be a standard pretrained bert-base-uncased model comprising of 110M parameters. We vary the weak model representation $h_w$ amongst different pretrained miniature BERT architectures [TCLT19]. Namely, we vary the hidden size $H \in \{128, 256, 512, 768\}$

and the number of transformer layers $L \in \{2, 4, 6, 8\}$ (the number of attention heads is always fixed at $H/64$). Pretrained weights for each of these is available online.

We perform the entire weak-to-strong supervision experiment identically as in Section 5.2—the gain-misfit plots are shown in Figure 5.

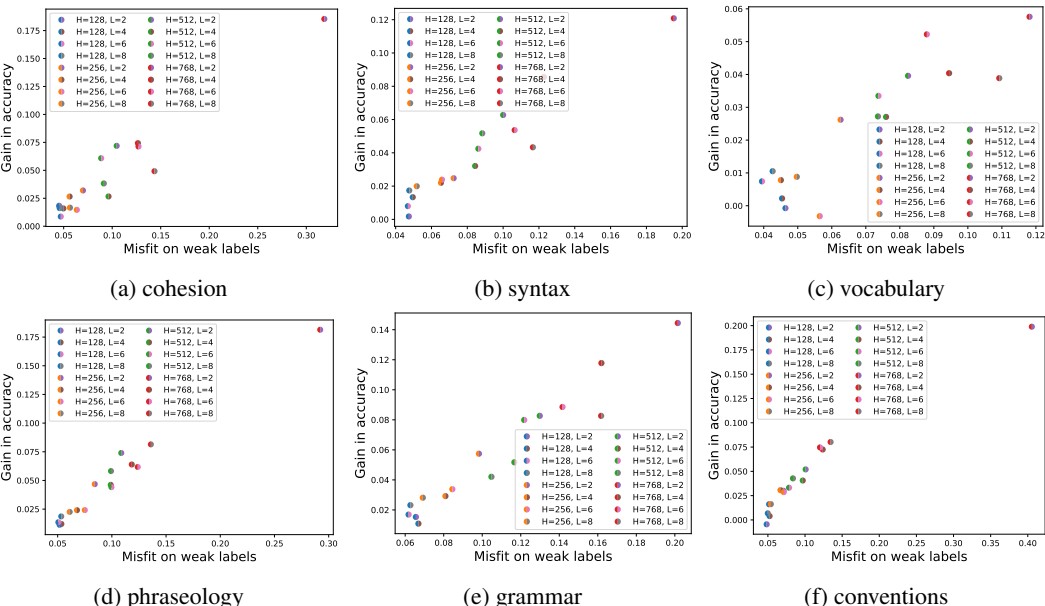

(a) cohesion  (b) syntax  (c) vocabulary

(d) phraseology  (e) grammar  (f) conventions

Figure 5: Results on the Essay Scoring dataset. Each plot corresponds to the task of predicting the score based on a different rubric.

## C.2  French Public Service Reviews

We also perform experiments on a dataset of public service reviews in French language collected from Google Maps and Trustpilot which accompanies the blog post of [R&D22] available at https://lajavaness.medium.com/regression-with-text-input-using-bert-and-transformers-71c155034b13. The task is to assign a score in 0-4 to the reviews, indicating whether it is negative or positive. The data source provides train and test splits.

We use pretrained models from the CamemBERT [MMS⁺20] and FlauBERT [LVF⁺20] family for this experiment. We fix $h_s$ to be camembert-large (335M parameters), and range $h_w$ amongst camembert-base, camembert-base-ccnet, camembert-base-wikipedia-4gb, camembert-base-oscar-4gb, camembert-base-ccnet-4gb, flaubert-small-cased, flaubert-base-cased and flaubert-base-uncased. The results can be seen in Figure 6.

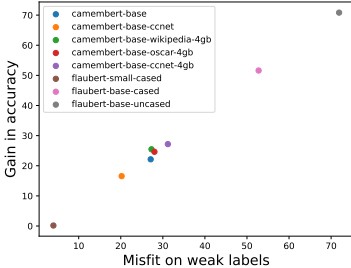

Figure 6: Results on the French Reviews dataset.

# D   Non-convex $\mathcal{F}_s$

Recall that in our Theorems 1 and 2, we crucially require the set $\mathcal{F}_s$ of finetuning tasks from which the strong model learns to be a convex set. Even in our experiments (Section 5), the set $\mathcal{F}_s$ is simply the set of linear functions (which is a convex set).

In this section, we instead consider the scenario where the set $\mathcal{F}_s$ is not convex. Concretely, consider the setting of the synthetic experiment in Figure 2b from Section 5.1, where we obtain the strong and weak representations via pretraining, the weak model is a 2-layer network and the strong model is an 8-layer network. Recall that the set $\mathcal{F}_s$ from which the strong and weak models learnt was the set of linear functions mapping $\mathbb{R}^{16} \to \mathbb{R}$ (this was also the ground-truth set from which the tasks were being generated from). Here instead, we consider the scenario where there is an additional non-linear activation $\phi$ following the linear map. Specifically, we consider the set $\mathcal{F}_s$ to be

$$\mathcal{F}_s = \{g : g = \phi \circ f, f \text{ is linear}\}.$$

The non-linear activation renders the set $\mathcal{F}_s$ to no longer be a convex set, and our theorems do not directly apply in this case. Thus, if we were to repeat the whole weak-to-strong generalization experiment for this $\mathcal{F}_s$, we have a priori no reason to expect the characterization of the gain in terms of the misfit. However, we observe that this characterization continues to hold for the three (popular) choices of $\phi$ that we tried, namely tanh, relu and sigmoid. The plots are shown in Figure 7. As we can see, even when the set $\mathcal{F}_s$ is clearly not convex, the gain is well-characterized by the misfit, suggesting that the result from Theorem 1 may hold even under more general settings.

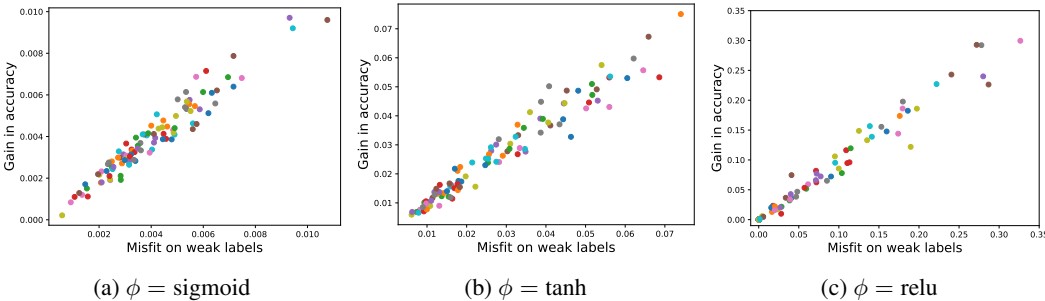

(a) $\phi = $ sigmoid        (b) $\phi = $ tanh        (c) $\phi = $ relu

Figure 7: Weak-to-strong generalization when $\mathcal{F}_s$ is a non-convex set.

# E   Discussion Regarding Extensions to Classification

While our weak-to-strong generalization theory is tailored to regression, we do believe that the qualitative conclusion of our work should also extend to other settings, including classification. In particular, as evidenced by numerous plots on student-supervisor disagreement in [BIK+23], the performance gain in classification ought also be (mathematically) characterized, at least to some extent, by the misfit between the weak and strong model; however, additional regularization terms in the objective (along with other error terms) seem to be necessary. This is supported by Figure 8 in [BIK+23], which shows that adding auxiliary losses in the objective helps reduce student-supervisor agreement as compared to naively finetuning (without any auxiliary loss) to the weak labels, and thereby improves the perfomance of the strong student model.

Of particular relevance here is also the study in Appendices E.1, E.2 in [BIK+23], which studies the *qualitative* structure of weak-to-strong disagreement in the setting of classification. Namely, the question studied there is: do different forms of disagreement between the strong and weak model (e.g., random errors vs correlated errors), for the same weak model accuracy, lead to differing weak-to-strong generalization? While this nuance arises in the setting of classification, it may be worth noting that in the setting of regression that we consider, projection onto the convex set only results in movement *towards* the true function, assuming realizability—in this sense, any misfit is only ever "in the correct direction", and its full quantitative benefit is realized. Thus, such a qualitative difference among different types of misfits does not manifest for us. Nevertheless, while it is true that in other settings, the nature of disagreement might matter, the general principle of decreasing student-supervisor imitation (alternatively, increasing misfit) to foster weak-to-strong generalization, either via auxiliary losses or otherwise, does constitute a significant theme in the work of [BIK+23].

## F    Implementation Details

All our synthetic experiments were run on a personal MacBook Pro 2021 with an Apple M1 Pro Chip (10 CPU cores) and no GPUs. All the gradient descent optimization procedures (pretraining tasks to obtain $h_w, h_s$, weak model finetuning, strong model finetuning on weak labels) used the Adam optimizer [KB14], with a batch size of 32, learning rate of $10^{-3}$ and 1000 epochs. We used the same random seed in all our synthetic experiments. All experiments completed execution within 2 hours.

The experiments on MolBERT used 2 GPUs with 8 GB memory on an internal GPU cluster. All experiments completed execution within 6 hours. We reuse much of the pretraining as well as finetuning codebase from the MolBERT repository (https://github.com/BenevolentAI/MolBERT/tree/main). The pretraining GuacaMol dataset as well as weights for a full-sized BERT model pretrained on this dataset can be downloaded from the repository. The MolBERT repository is under the MIT license, and the ChemBench repository is under the Python Packaging Authority license.

