# OpenReview forum: "Quantifying the Gain in Weak-to-Strong Generalization"
_NeurIPS.cc/2024/Conference — NeurIPS 2024 poster_

### Official Review · Reviewer_322n · 2024-07-11

**Soundness:** 4
**Presentation:** 4
**Contribution:** 3
**Rating:** 7
**Confidence:** 4

**Summary:**

This paper studies the phenomena of weak-to-strong generalisation (WTSG) from a theoretical angle, aiming to explain why the phenomena occurs. The paper's main theoretical results show that, in a regression setting with a convex function class, the decrease in MSE from the weak to weak2strong model is bounded below by the error of the weak2strong model on the weak labels (the "misfit"). The proof of the theorem relies on the pythagoreon theorem for projectsions onto a convex set. The paper then performs synthetic (simulated guassian data) and realistic (molecular property prediction) experiments to test out the theoretical prediction, and find that the decrease in MSE is roughly linearly proportional to the misfit on the weak labels in both settings. They show that this relationship can be used to select which weak2strong model to choose (the one with the highest misfit) correctly. They finally show that in a low-sample regime, "weak" models aren't necessarily smaller ones, as larger models may overfit to the small number of samples and hence perform worse. In this setting, their predicted relationship continues to hold, but with the small model being the strong model.

**Strengths:**

The weak-to-strong generalisation (WTSG) phenomena is an important one to study and understand. The theoreticaly explanation put forth in this paper is easy to understand, original, and produces novel insight into why WTSG happens. The confirmation of those findings empirically, including in a more realistic setting, strengthens the work's quality and significance. The paper is generally well-written and clear and easy to read.

**Weaknesses:**

It's unclear why the authors chose the regression setting, when the original work and likely scenario of use is classification setting. I think this difference from the original work should be made clearer.

The empirical confirmation in realistic data is limited to one setting, which is an unfamiliar dataset to me and likely most of the LLM and NLP community. It would be beneficial to demonstrate the phenomena across more settings, especially more standard ones.

Giving some intuition as to *why* the theoretical result holds from the proof would be beneficial. In particular, pointing out that the inequality achieved relies on the distance measure not being a metric, otherwise the triangle inequality for metrics would force the inequality to be the other way round, if my understanding is correct?

It's likely that in the setting from the original paper, the weak labelling is within the function class of fine-tuned strong models (where fine-tuning is over all parameters), at least on a naïve interpretation. It would be beneficial to discuss whether the authors think their theoretical results is the reason why WTSG happens in this setting as well (given the assumptions of their theory break), or whether they think WTSG happens in that scenario for a different reason.

### Summary

In general, I'm in favour of the paper getting accepted, and am giving it an accept (7). I'd be willing to raise my score if additional empirical justification of theoretical claims was made in more varied settings.

**Questions:**

-

**Limitations:**

The authors have adequately discussed the limitations of the work.

---

> ### Author Rebuttal · Authors · 2024-08-07
>
> Thank you so much for reading our paper, and for your review and comments. We are really glad that you like our work!
>
> With regards to the points you bring up:
>
> > It's unclear why the authors chose the regression setting, when the original work and likely scenario of use is classification setting. I think this difference from the original work should be made clearer.
>
> Regression with the least squares loss, while arguably being one of the most basic tasks in statistics and machine learning, also turns out to be a setting where we can extract the essence of the weak-to-strong generalization phenomenon in an intuitive, geometric manner — as we show, the phenomenon in this setting effectively boils down to the Pythagorean theorem for projection onto convex sets (a well-studied and standard fact from convex analysis). We do believe that a similar intuition and explanation (in terms of “projections” onto “convex” sets) may also apply to the classification setting (with say the binary cross-entropy loss). It is also worth mentioning that the theory in several previous studies [1,2] on out-of-distribution finetuning is also focused on regression. Nevertheless, we will emphasize more on the difference in our setting (regression vs classification) in the next revision.
>
> [1] Ananya Kumar, Aditi Raghunathan, Robbie Jones, Tengyu Ma, and Percy Liang. Fine tuning can distort pretrained features and underperform out-of-distribution.
>
> [2] Yoonho Lee, Annie S Chen, Fahim Tajwar, Ananya Kumar, Huaxiu Yao, Percy Liang, and Chelsea Finn. Surgical fine-tuning improves adaptation to distribution shifts.
>
> > The empirical confirmation in realistic data is limited to one setting, which is an unfamiliar dataset to me and likely most of the LLM and NLP community. It would be beneficial to demonstrate the phenomena across more settings, especially more standard ones.
>
> The molecular prediction task seemed natural for our setting and also stood out to be a standardized task [1] in computational chemistry for regression over sequential data, with a tractable pre-training dataset, and a good variety of fine tuning tasks to demonstrate the applicability of our results. Additionally, the community had also previously developed specialized transformer architectures for these tasks [2]. That is why we decided to go with this dataset.
>
> As can also be seen, for example, in Burns et al., 23, a major bulk of NLP/LLM tasks seem to be tailored to classification/generation; nevertheless, we would be happy to include results on a suitable NLP regression task (suggestions welcome!) in the next version.
>
> [1] Zhenqin Wu, Bharath Ramsundar, Evan N Feinberg, Joseph Gomes, Caleb Geniesse, Aneesh S Pappu, Karl Leswing, and Vijay Pande. Moleculenet: a benchmark for molecular machine learning
>
> [2] Benedek Fabian, Thomas Edlich, Héléna Gaspar, Marwin Segler, Joshua Meyers, Marco Fiscato, and Mohamed Ahmed. Molecular representation learning with language models and domain-relevant auxiliary tasks.
>
> > Giving some intuition as to why the theoretical result holds from the proof would be beneficial. In particular, pointing out that the inequality achieved relies on the distance measure not being a metric, otherwise the triangle inequality for metrics would force the inequality to be the other way round, if my understanding is correct?
>
> You are indeed right that if the distance measure (average squared distance) was a metric, we would get the reversed inequality. An intuitive reason for the inequality being the in other direction is the textbook cosine law on the sides of a triangle: $a^2 = b^2 + c^2 - 2bc \cos(\theta)$, where $\theta$ is the angle between sides $b$ and $c$. Convexity guarantees that this angle is obtuse, which makes the cosine negative. Hence, $c^2 \le a^2 - b^2$. Here, $a^2$ is the weak model loss, $b^2$ is the weak-to-strong misfit, and $c^2$ is the strong model loss.
>
> > It's likely that in the setting from the original paper, the weak labelling is within the function class of fine-tuned strong models (where fine-tuning is over all parameters), at least on a naïve interpretation. It would be beneficial to discuss whether the authors think their theoretical results is the reason why WTSG happens in this setting as well (given the assumptions of their theory break), or whether they think WTSG happens in that scenario for a different reason.
>
> This is a great question, as in such a case, our theory (which works in the setting of regression) would indicate that no weak-to-strong generalization should be exhibited. This is also what the authors of Burns et al, 23 refer to as *perfect student-supervisor agreement/imitation*. To mitigate this (Section 4.3.2 in Burns et al., 23), notice that the authors of that paper suggest incorporating an auxiliary "confidence" loss in their finetuning optimization objective to enable weak-to-strong generalization—this serves as a regularizer, and avoids the student (strong) model to naively overfit the weak labels. Note however, that our theory heavily uses the fact that the minimizer of the (unregularized) mean squared error is in fact the *projection* of the weak model onto the convex set—this is no longer true with a regularization term. However, there is a natural duality between regularized objectives and constrained optimization problems. An interesting future direction then would be to characterize the regularization terms that constrain the space of optimization to still contain the target function, but exclude the weak model itself, so as to provably mitigate student-supervisor imitation.
>
> ---
>
> Please let us know if we can help answer any other questions!

---

> > ### Comment · Reviewer_322n · 2024-08-10
> >
> > Thanks for your clarification and response.
> >
> > In response to your last point, while Burns et al do propose the auxiliary confidence loss, even without this loss (i.e. naïve WTSG), the w2s model outperforms the weak model, which is anti-predicted by your theory (which as you note would predict no performance boost on top of the weak model). Ignoring the auxiliary confidence loss entirely Burns et al's results still don't match your theory, so I'll ask again whether you have any ideas as to why that is the case: Do you think your theoretical results are the reason why WTSG happens in this setting as well (given the assumptions of your theory break), or do you think WTSG happens in that scenario for a different reason?
> >
> > I agree that it would be interesting to extend your theory to explaining the auxiliary confidence loss' boost in performance, but you would first need to ensure it explains the naïve WTSG effect in the full fine-tuning regime, as it currently does not.
> >
> > Regardless, I am still happy to recommend acceptance of the paper, and will maintain my score.

---

> ### Author Response · Authors · 2024-08-10
> **Response to comment**
>
> Thank you for your comment. We do believe that the performance gains via WTSG in classification settings ought to also be characterized to some extent by the misfit between weak and strong models; however, other terms also seem necessary (capturing, among other factors, the case where the strong model can represent the weak model). Indeed, the gains in accuracy with naive finetuning without auxiliary losses in the classification settings considered by Burns et al. 23 already suggest that such terms should be necessary, and that the phenomenon of accuracy gain via WTSG in classification is more nuanced than in regression (e.g., not only the quantity, but even the *quality* of misfit can matter here). Thus, our theory does not directly explain the gains in performance in these cases, and a different analysis seems to be required.
>
> As for initial thoughts about a theoretical explanation for gains in classification: the phenomenon in classification that we want to capture is the following: the weak model is presenting the strong model with labeled data; however, the labels are really only "pseudolabels", and not true labels. The strong model, while learning from the weak model, is nevertheless still *rectifying* the wrong "pseudolabels". The self-training framework of [1], where the student and teacher are the same model, does indeed capture the case where the student model can in principle fully represent the weak model. Under certain assumptions (like expansion and separation), and along with consistency losses, this theory does explain performance gains of the self-trained student model, and thus seems to be a promising avenue to explain gains in WTSG as well. It would be interesting to see if the high-level analysis in our work (projection onto the space of strong models that are already implicitly biased towards the ground truth) could be combined with such an analysis to explain performance gains, even when no consistency losses are involved. Again, all these are really fascinating directions for future study!
>
> [1] Colin Wei, Kendrick Shen, Yining Chen, and Tengyu Ma. Theoretical analysis of self-training with deep networks on unlabeled data.

---

> > ### Comment · Reviewer_322n · 2024-08-12
> >
> > Thanks for the response and discussion.
> >
> > My point was that main difference between your analysis and Burns et al. is not classification vs regression, but that your analysis assumes probing pretrained models, as opposed to fine-tuning all the weights. If you were to perform full fine-tuning in the regression setting then I would expect you would still see WTSG, but your theory would not predict that, as I think full fine-tuning of the strong model should be able to completely fit the labels of the weaker model (i.e. weak model function is inside the strong model function class), as it is a bigger model. Your theory would not predict WTSG here I think? If you ran WTSG for regression with full fine-tuning, what would you predict would happen, and how might you extend your theory to cover that case?

---

> > > ### Author Response · Authors · 2024-08-12
> > > **Response to comment**
> > >
> > > Thank you so much for the clarification. We really appreciate your time engaging in this discussion, and acknowledge that our analysis assumes linear probing of pretrained models, whereas the gains in the classification settings of Burns et al. hold even when all the parameters of the strong model are finetuned. Here are a few thoughts:
> > >
> > > 1) You are right that if we allow finetuning of all the parameters of the strong model in the weak supervision stage, the weak model function is inside the strong model function class. However, we would like to note that in this case, the class of strong model functions ($f \circ h_s$ where both $f$, $h_s$ are free) is no longer a convex set---thus, our theory doesn't extend to this setting, and doesn't exactly predict whether we should/should not see WTSG.
> > >
> > > 2) Nevertheless, we ran a quick experiment on synthetic data (as in Figure 2(a), 2(b) of the paper), where we finetuned all the parameters in the strong model in the weak supervision stage. We would like to remark that we didn't see a clear WTSG trend in these experiments. It still might be true that on large-scale, real data, WTSG is observed in the regression setting, even if we were to finetune all the strong model parameters (like what we see in Burns et al.). If this is so, it would be interesting to characterize why this is happening, given that the underlying convexity assumption from our theory is broken (e.g., does the number of finetuning examples matter in the weak supervision stage?) On the other hand, if no clear WTSG is seen even in large scale experiments (like in the synthetic data), this would also be very interesting and suggest that 1) the conclusion of our analysis may be extendable to non-convex spaces of strong model functions, and 2) even further differences in WTSG between classification and regression.

---

### Official Review · Reviewer_8aZZ · 2024-07-14

**Soundness:** 3
**Presentation:** 1
**Contribution:** 2
**Rating:** 5
**Confidence:** 4

**Summary:**

The paper provides a geometric perspective on weak-to-strong generalization [[Burns et al., 23](https://arxiv.org/abs/2312.09390)]. Specifically, the authors show that if the set of strong model-representable functions the following holds:
$$MSE(\phi^*, \phi^{ws}) \le MSE(\phi^*, \phi^w) - MSE(\phi^w, \phi^{ws}),$$

where $\phi^*$ is the ground truth labeling function, $phi^{w}$ is the the weak model labeling function and $\phi^{ws}$ is the weak-to-strong model labeling function. By labeling function I mean a mapping from inputs $x$ to real-valued labels $y$.

The authors use this result to argue that the gain in MSE loss in weak-to-strong generalization is at least equal to the _misfit_ term $MSE(\phi^w, \phi^{ws})$. There are some experiments verifying this result in practice.

**Strengths:**

1. The core result is in fact very simple and intuitive: it's a geometric argument about a projection on a convex set.

2. To the best of my knowledge, this main result was not reported previously in the context of weak-to-strong generalization. It provides a toy model where it's trivial to see that weak-to-strong generalization will hold.

3. The authors show some simple experiments where their theory is applicable: synthetic regression tasks and molecular property prediction.

**Weaknesses:**

W1. My main concern with the paper is that in my opinion the presentation is very confusing. Specifically, the core result is very simple, but it is presented in a way that took me a while to understand.

My first issue is that the authors use notation $d_{\mathcal{P}}$ to denote the mean-squared distance, i.e. $\mathbb{E}(f(x) - g(x))^2$ and refer to it as distance (line 133). If $d_{\mathcal{P}}$ was in fact a distance, then Eq. (2) is the inverse of the triangular inequality. This confused me for a while. Moreover, Fig 1 shows a triangle with sides labeled as $A$, $B$, $C$, and states $C \ge A + B$. The resolution is that $d_{\mathcal{P}}$ is a square of a distance, not a distance.

Then, Theorem 1 is just a combination of (1) law of cosines and (2) and the fact that the angle between the vectors connecting a point to its projection on a convex set, and a vector connecting its projection to any point in the convex set is $\ge 90$ degrees.

The proof and the statement of the theorem are in my opinion significantly complicated by the explicit use of representations $h$ throughout. I don't understand what value they add: throughout the paper, the authors always use the same compositions e.g. $f^s \circ h^s$. Denoting the entire mapping with one letter would simplify the presentation. Currently, the authors also state that they are "employing a representation-theoretic perspective" (line 308), but I don't think there is anything added by the parameterization of the model as a function on top of some representations. Possibly the only thing is Claim 3, but it doesn't affect other parts of the paper much.

W2. The result of the paper holds specifically for the MSE loss. Indeed, if $d_{\mathcal{P}}$ is a distance metric, then Eq.2 is the opposite of the triangle inequality and doesn't hold. Moreover, for the qualitative conclusion
> Thus, the 148 inequality directly says that the weakly-supervised strong model improves over the weak model by (at least) an amount equal to the misfit.

clearly doesn't hold for the classification settings considered by [Burns et al., 23]: it is possible for the strong model to differ from the weak model on datapoints where the weak model is correct, which would not make it more accurate with respect to the ground truth labels.

W3. In fact, [Burns et al., 23] provide a relevant discussion of _simulation difficulty_ in Appendix E (especially E.1). They also note that in order for the strong model to improve upon the weak supervisor, the strong model should be unable to simulate the weak labels well. A related result is also reported in Fig 8 of that paper. I think it would be good for the authors to comment on how their results connect to these results.

W4. The practical proposition of using the weak model with the highest misfit will not work without further assumptions. This idea is again quite related to the result in Appendix E.1 of [Burns et al., 23]: if the errors of the weak model are not simulatable by the strong model, but the signal is easy to simulate, then we will get very good weak-to-strong generalization. But in general, high weak-to-strong disagreement will not always imply good weak-to-strong generalization.

Even under the assumptions of convexity etc of theorem 1, I believe it is not true that for a given $d_{\mathcal{P}}(\phi^*, \phi^w)$ higher $d_{\mathcal{P}}(\phi^w, \phi^{ws})$ always lead to lower $d_{\mathcal{P}}(\phi^*, \phi^{ws})$. Is that correct?

**Questions:**

Please comment on W1-W4.

**Limitations:**

Limitations are adequately discussed.

---

> ### Author Rebuttal · Authors · 2024-08-07
>
> Thank you so much for reading our paper, and for your comments. We address your concerns ahead:
> > W1..presentation very confusing..notation $d_P$ to denote the mean-squared distance..
>
> The only reason we introduced the notation $d_P(f,g)$ was for ease of reading: it is convenient to have some notation for the long-form term $E_P(f(x)-g(x))^2$. We explicitly state in line 133 that $d_P$ is the “avg squared distance wrt $P$" (this is already not a valid "distance" in the context of metric spaces, which indeed must satisfy the triangle inequality).
>
> We want to assert that, if anything, we have tried to be as explicit and upfront about the simplicity and intuitive nature of our result as possible. There was no intent to make anything confusing. Nevertheless, we will add a line clarifying that $d_P$ does not satisfy the triangle inequality. We will also consider changing $d_P$ to $MSE$,just as you use in your review, and denoting the sides of the triangle as $A^2, B^2, C^2$ in Fig 1.
> > Thm 1 just a combination of law of cosines and convexity
>
> This is indeed the right intuition; formalizing it to the space of functions requires a proof, which is what we include. To us, the fact that weak-to-strong generalization (WTSG) (a seemingly complex, and increasingly important phenomenon going ahead) in the concrete setting of regression can be directly traced to the Pythagorean thm for convex sets (a std fact in convex analysis) is both surprising and satisfying. The main contribution of our work is in modeling this phenomenon and formalizing the question. In this capacity, we view simple explanations as a strength and not a weakness, as they help us understand the system better.
> > ...proof complicated by explicit use of representations throughout
>
> The only reason we frame our results as “finetuning on top of representations” is because this is predominantly the language used for understanding AI models today. Your suggestion about clubbing $f_s \circ h_s$ to $\phi_s$ is well-taken; however, if we simply denote the composition by a single function, the result appears more abstract, and the link to WTSG is obfuscated. If anything, our intention was for the reader to appreciate that the working of these models can be elicited at a level where one can instantiate standard mathematical tools.
> > W2. result of the paper holds only for the MSE loss.
>
> Indeed, and this is something we repeatedly make clear to be the scope of the paper (line 67, 133, 309..). This is a first step towards understanding the phenomenon of WTSG, and extending to other settings is an interesting future direction.
> > ...qualitative conclusion clearly doesn't hold for classification...possible for strong model to differ from weak model on points where weak model is correct
>
> It is a priori not clear that the qualitative conclusion of our work (that the gain in WTSG is effectively characterized by the WTS misfit) does not extend to settings beyond regression. It may very well be possible that an inequality like Eq 2 (with suitable error terms) holds in the classification setting, with $d_P$ measuring (say) the avg binary cross-entropy loss. Moreover, in such a case, it may still be possible that the strong model makes a mistake on a point where the weak model does not; the overall gain in performance will likely stem from the fact that the probability under the data distribution on such points is low, whereas the probability on points where the strong model improves is high. In any case, this is an interesting direction beyond the scope of the present work.
> > W3...Appendix E in Burns et al., 23, where they note that the strong model improves upon the weak supervisor when it is unable to simulate the weak labels
>
> This is precisely what our main theorem is also saying! Any strong model that suffers a large misfit error when trained (without auxiliary losses) on weak labels (i.e., it is unable to simulate the weak labels well) exhibits non-trivial WTSG! Relatedly, Fig 8 in their paper shows that adding auxiliary losses in the objective helps reduce student-supervisor agreement (alternatively, increase misfit), and thereby improve WTSG.
> > W4...in general, high WTS disagreement will not always imply WTSG
>
> The study in Appendix E.1,E.2 in Burns et al. 23 is more to do with the *qualitative* structure of WTS disagreement in the setting of classification. Namely, they ask: do different forms of disagreement (e.g., random errors vs correlated errors), for the same weak model accuracy, lead to differing WTSG?
>
> While this nuance arises in the setting of classification, in the setting of regression that we consider, projection onto the convex set only results in movement *towards the true function* assuming realizability---in this sense, any misfit is "in the correct direction”, and its full quantitative benefit is realized. Thus, the qualitative difference among different forms of misfits does not manifest for us. While it is true that in other settings, the nature of disagreement might matter, the general principle of decreasing student-supervisor imitation (alternatively, increasing misfit) to foster WTSG, either via auxiliary losses or otherwise, does constitute a significant theme in the work of Burns et al., 23.
> > Even under the assumptions of convexity etc of Thm 1...
>
> As our theorem asserts, for the regression setting under the squared loss and with a convex space of finetuning functions, it is (mathematically) true that for a given $d_P(\phi^*, \phi^w)$, a higher $d_P(\phi^w, \phi^{ws})$ (where $\phi^{ws}$ is the projection/minimizer of loss) will indeed lead to lower $d_P(\phi^*, \phi^{ws})$ (this is verbatim the inequality). In fact, this is also (nearly exactly) corroborated by all our experiments.
>
> ---
> We will definitely add a summary of the discussion above, in relation to Appendix E in Burns et al., 23 in the final revision. We hope that our response helps address your concerns. Please let us know if we can answer any more questions.

---

> > ### Author Response · Authors · 2024-08-12
> > **Checking in....**
> >
> > Thank you so much again for taking the time to read our manuscript! As the deadline for the discussion period is approaching, we would really appreciate hearing from you, and ask if any further clarifications are required!

---

### Official Review · Reviewer_oUtk · 2024-07-15

**Soundness:** 3
**Presentation:** 3
**Contribution:** 3
**Rating:** 7
**Confidence:** 5

**Summary:**

This paper provides bounds for weak-to-strong generalization, where a strong student model is trained on the labels produced by a weaker teacher model. The authors prove that in a regression setup, under certain assumptions, the strong model gains over the weak model's accuracy by an amount equal to the *disagreement* between the strong and weak model. This gives a natural generalization bound for weakly-supervised regression models and also gives a selection criterion for which weak model to choose in practice, which the experiments show leads to good empirical performance.

**Strengths:**

- Proves a clean and intuitive theory for weak-to-strong regression.

- Empirical results showing that the proposed bounds are tight (in fact, almost exact).

**Weaknesses:**

- The results of WSCM20 are not properly contextualized. Their analysis is *not* limited to a self-training scenario and applies for any student model learning from an arbitrary teacher, including a student that is more powerful than the teacher.

- The paper is missing a discussion of and citations to relevant work in other semi- or un-supervised settings that bound generalization error in terms of the disagreement between two classifiers, such as [1], [2], and especially [3].

[1] https://arxiv.org/abs/1708.04403
[2] https://arxiv.org/abs/1901.04215
[3] https://papers.nips.cc/paper_files/paper/2001/file/4c144c47ecba6f8318128703ca9e2601-Paper.pdf

**Questions:**

- L71 "Next, we imagine that the weak model sees data labeled by the target function $f^* \circ h^*$, and after finetuning, learns some arbitrary function $f_w \circ h_w$."

    Is this assumption really necessary? Can't we just assume we are given a weak predictor with some error rate? This limits the setup significantly to weak teacher models that are fine-tuned on ground-truth data.

- What happens in the current theory if technically the strong model hypothesis class $\mathcal{F}_s$ technically contains a function that can exactly fit the weak classifier $f_w \circ h_w$, but due to regularization this doesn't occur? This is often the case in practice, where regularization terms are required to avoid overfitting to the weak labels. Can the theory be modified to include a regularized version of $\mathcal{F}_s$?

**Limitations:**

The paper already discusses what are in my view its main limitations: (1) it only applies to regression and (2) it only applies to settings where the strong model hypothesis class $\mathcal{F}_s$ is convex.

---

> ### Author Rebuttal · Authors · 2024-08-07
>
> Thank you so much for reading our paper, and for your review and comments. We are really glad that you like our work!
>
> With regards to the points you bring up:
>
> > The results of WSCM20 are not properly contextualized. Their analysis is not limited to a self-training scenario and applies for any student model learning from an arbitrary teacher, including a student that is more powerful than the teacher.
>
> Thank you for pointing this out. We will clarify that the analysis of [WSCM20] applies to arbitrary student-teacher settings (albeit under expansion assumptions and a consistency loss) in the updated version!
>
> > missing a discussion of and citations to relevant work in other semi- or un-supervised settings that bound generalization error in terms of the disagreement between two classifiers
>
> Thank you for pointing out these references on disagreement-based analyses. We will be sure to include them in the revision.
>
> > L71 "Next, we imagine that the weak model sees data labeled by the target function $f^* \circ h^*$, and after finetuning, learns some arbitrary function $f_w \circ h_w$." Is this assumption really necessary? Can't we just assume we are given a weak predictor with some error rate? This limits the setup significantly to weak teacher models that are fine-tuned on ground-truth data.
>
> You are right, our results broadly hold under access to any weak predictor. Note that we do mention in our statement of Theorem 1 that $h_w$ is any weak representation, and $f_w$ is any (arbitrary) predictor (so that $f_w \circ h_w$ is any arbitrary predictor). Nevertheless, we will rephrase the sentence in the introduction to make it clear that the weak predictor can really be any arbitrary predictor.
>
> > What happens in the current theory if technically the strong model hypothesis class $\mathcal{F}_s$  technically contains a function that can exactly fit the weak classifier $f_w \circ h_w$, but due to regularization this doesn't occur? This is often the case in practice, where regularization terms are required to avoid overfitting to the weak labels. Can the theory be modified to include a regularized version of $\mathcal{F}_S$?
>
> This is a great question. In the setting considered in our paper, if $f_w \circ h_w$ can be represented exactly as $f_s \circ h_s$ for some $f_s \in \mathcal{F}_s$, and we were to perform naive finetuning without any regularization, the strong model will exactly converge to $f_w \circ h_w$, and we will see no weak-to-strong generalization. As also suggested in the work of Burns et al., 23, auxiliary “confidence” losses serve as regularizers to mitigate this phenomenon. Note however, that our theory heavily uses the fact that the minimizer of the (unregularized) mean squared error is in fact the *projection* of the weak model onto the convex set—this is no longer true with a regularization term. However, there is a natural duality between regularized objectives and constrained optimization problems. An interesting future direction then would be to characterize the regularization terms that constrain the space of optimization to still contain the target function, but exclude the weak model itself, so as to provably mitigate student-supervisor imitation.
>
> ---
>
> Please let us know if we can help answer any other questions!

---

> > ### Author Response · Authors · 2024-08-12
> > **Checking in....**
> >
> > Thank you so much again for taking the time to read our manuscript! As the deadline for the discussion period is approaching, we would really appreciate hearing from you, and ask if any further clarifications are required!

---

### Decision · Program_Chairs · 2024-09-25

**Decision:**

Accept (poster)

**Comment:**

**Paper summary**

The paper contributes bounds to characterize weak-to-strong generalization (WTSG) in a regression setting. In this setup, a strong student model is trained on labels produced by a weak teacher. The teacher is trained on the true observed labels. The main result is (slightly rewritten for simplicity)

$$\mathrm{MSE}(\phi_w, \phi_{sw}) \le \mathrm{MSE}(\phi^*, \phi_w) - \mathrm{MSE}(\phi^*, \phi_{sw}),$$

where $\phi^*$ is the ground-truth data generating function, $\phi_w$ is the weak teacher, and $\phi_{sw}$ is the strong model trained on labels produced from the weak teacher. The inequality says that the reduction in error from the strong model (right-hand side) is at least the difference between the weak and the strong model’s predictions (left-hand side), which the authors refer to as “misfit”. Experiments on both synthetic and real data verify the correctness of the bounds.

**Review**

The paper presents clean theoretical bounds to characterize WTSG in a regression setting (oUtk). The results produce novel insights into why WTSG happens (322n), and are intuitive, relying on a geometric argument about a projection onto a convex set (8aZZ). Experimental  results confirm that the bounds are tight (oUtk, 8aZZ).

The rebuttal from the authors was strong and to the point, which eventually convinced the reviewers. Several concerns were raised, including confusing presentation (mainly about the notation), insufficient discussion on relevant results from Burns et al., 2023, limited experiment on real data (only one dataset), and finally limited study of only the MSE loss (8aZZ, 322n). All have been addressed sufficiently. The authors promised to include more experiments on NLP regression tasks.

Of all the concerns, the last point (limited study on only the MSE loss) appears to be the most important. The authors have made it clear in the paper that the results only hold for this setting. In AC’s view, explaining WTSG in, say, classification tasks can be more nuanced than in regression. The cleanliness of the theoretical results and the nice geometric argument are made possible because of the squared loss. Despite the limited setting, explaining WTSG intuitively is of value.


Recommendation: accept.

Please include more experiments on NLP regression tasks in the next revision.